# Overall and cause-specific hospitalisation and death after COVID-19 hospitalisation in England: A cohort study using linked primary care, secondary care, and death registration data in the OpenSAFELY platform

Krishnan Bhaskaran[1][�]*, Christopher T. Rentsch[1][�], George Hickman[2][�], William J. Hulme[2][�], Anna Schultze[1], Helen J. Curtis[2], Kevin Wing[1], Charlotte Warren-Gash[1], Laurie Tomlinson[1], Chris J. Bates[3], Rohini Mathur[1], Brian MacKenna[2], Viyaasan Mahalingasivam[1], Angel Wong[1], Alex J. Walker[2], Caroline E. Morton[2], Daniel Grint[1], Amir Mehrkar[2], Rosalind M. Eggo[1], Peter Inglesby[2], Ian J. Douglas[1], Helen I. McDonald[1], Jonathan Cockburn[3], Elizabeth J. Williamson[1], David Evans[2], John Parry[3], Frank Hester[3], Sam Harper[3], Stephen JW Evans[1], Sebastian Bacon[2], Liam Smeeth[1][‡], Ben Goldacre[2][‡]

**1** Faculty of Epidemiology and Population Health, London School of Hygiene and Tropical Medicine, London, United Kingdom, **2** The DataLab, Nuffield Department of Primary Care Health Sciences, University of Oxford, Oxford, United Kingdom, **3** TPP, TPP House, Horsforth, Leeds, United Kingdom

he These authors contributed equally to this work.
‡ These authors are joint senior authors on this work.
* krishnan.bhaskaran@lshtm.ac.uk

## Abstract

### Background

There is concern about medium to long-term adverse outcomes following acute Coronavirus Disease 2019 (COVID-19), but little relevant evidence exists. We aimed to investigate whether risks of hospital admission and death, overall and by specific cause, are raised following discharge from a COVID-19 hospitalisation.

### Methods and findings

With the approval of NHS-England, we conducted a cohort study, using linked primary care and hospital data in OpenSAFELY to compare risks of hospital admission and death, overall and by specific cause, between people discharged from COVID-19 hospitalisation (February to December 2020) and surviving at least 1 week, and (i) demographically matched controls from the 2019 general population; and (ii) people discharged from influenza hospitalisation in 2017 to 2019. We used Cox regression adjusted for age, sex, ethnicity, obesity, smoking status, deprivation, and comorbidities considered potential risk factors for severe COVID-19 outcomes.

We included 24,673 postdischarge COVID-19 patients, 123,362 general population controls, and 16,058 influenza controls, followed for ≤315 days. COVID-19 patients had median

**Data Availability Statement:** All code is shared openly for review and re-use under MIT open license (https://github.com/opensafely/post-admission-admissions-research). Detailed pseudonymised patient data is potentially re-identifiable and therefore not shared under our agreed information governance and ethics approvals.

**Funding:** This work was jointly funded by UKRI, NIHR and Asthma UK-BLF [COV0076; MR/V015737/] and the Longitudinal Health and Wellbeing strand of the National Core Studies programme. The OpenSAFELY data science platform is funded by the Wellcome Trust. TPP provided technical expertise and infrastructure within their data centre pro bono in the context of a national emergency. KB holds a Senior Research Fellowship from Wellcome (220283/Z/20/Z). RME is funded by HDR UK (grant: MR/S003975/1) and MRC (grant: MC_PC 19065). RM holds a Sir Henry Wellcome fellowship funded by Wellcome. The funders had no role in study design, data collection and analysis, decision to publish, or preparation of the manuscript.

**Competing interests:** I have read the journal's policy and the authors of this manuscript have the following competing interests. The authors declare that: AS is employed by LSHTM on a fellowship sponsored by GSK. CWG is supported by a Wellcome Intermediate Clinical Fellowship (201440/Z/16/Z), and also holds grants from the Alzheimer's Society, the British Heart Foundation and the Rosetrees Trust for unrelated work. RM has received consulting fees from AMGEN unrelated to the submitted work. ID has received grants from and holds shares in GSK. HIM is funded by the National Institute for Health Research (NIHR) Health Protection Research Unit (HPRU) in Vaccines and Immunisation, a partnership between Public Health England and the London School of Hygiene & Tropical Medicine. JP is an employee of TPP (Leeds) Ltd who own SystmOne. BG has received research funding from the Laura and John Arnold Foundation, the NHS National Institute for Health Research (NIHR), the NIHR School of Primary Care Research, the NIHR Oxford Biomedical Research Centre, the Mohn-Westlake Foundation, NIHR Applied Research Collaboration Oxford and Thames Valley, the Wellcome Trust, the Good Thinking Foundation, Health Data Research UK (HDRUK), the Health Foundation, and the World Health Organisation; he also receives personal income from speaking and writing for lay audiences on the misuse of science.

**Abbreviations:** aHR, adjusted hazard ratio; BMI, body mass index; COVID-19, Coronavirus Disease

age of 66 years, 13,733 (56%) were male, and 19,061 (77%) were of white ethnicity. Overall risk of hospitalisation or death (30,968 events) was higher in the COVID-19 group than general population controls (fully adjusted hazard ratio [aHR] 2.22, 2.14 to 2.30, $p < 0.001$) but slightly lower than the influenza group (aHR 0.95, 0.91 to 0.98, $p = 0.004$). All-cause mortality (7,439 events) was highest in the COVID-19 group (aHR 4.82, 4.48 to 5.19 versus general population controls [$p < 0.001$] and 1.74, 1.61 to 1.88 versus influenza controls [$p < 0.001$]). Risks for cause-specific outcomes were higher in COVID-19 survivors than in general population controls and largely similar or lower in COVID-19 compared with influenza patients. However, COVID-19 patients were more likely than influenza patients to be readmitted or die due to their initial infection or other lower respiratory tract infection (aHR 1.37, 1.22 to 1.54, $p < 0.001$) and to experience mental health or cognitive-related admission or death (aHR 1.37, 1.02 to 1.84, $p = 0.039$); in particular, COVID-19 survivors with preexisting dementia had higher risk of dementia hospitalisation or death (age- and sex-adjusted HR 2.47, 1.37 to 4.44, $p = 0.002$). Limitations of our study were that reasons for hospitalisation or death may have been misclassified in some cases due to inconsistent use of codes, and we did not have data to distinguish COVID-19 variants.

## Conclusions

In this study, we observed that people discharged from a COVID-19 hospital admission had markedly higher risks for rehospitalisation and death than the general population, suggesting a substantial extra burden on healthcare. Most risks were similar to those observed after influenza hospitalisations, but COVID-19 patients had higher risks of all-cause mortality, readmission or death due to the initial infection, and dementia death, highlighting the importance of postdischarge monitoring.

## Author summary

### Why was this study done?

- Early studies have suggested that some people infected with SARS-CoV-2 may be at risk of developing health problems in the months after their initial infection. Given high rates of infection in many countries, this is a significant public health concern, but there is currently limited evidence to inform policy.

- The aim of this study was to systematically quantify the extent to which people who have been in hospital with COVID-19 may be at higher risk of dying or being readmitted to hospital, either overall or for specific illnesses, compared with people in the general population, and people who have been hospitalised with influenza.

### What did the researchers do and find?

- We used a cohort study design to compare risks of hospitalisation and death, overall and for a range of specific causes, between people who had hospitalised with COVID-19 ($n = 24,673$), people with similar demographic characteristics in the 2019 general

2019; HR, hazard ratio; ICD, International Classification of Diseases; LRTI, lower respiratory tract infection; ONS, Office of National Statistics; SARS-CoV-2, Severe Acute Respiratory Syndrome Coronavirus 2; STP, Sustainability and Transformation Plans; SUS, Secondary Uses Service.

population ($n$ = 123,362), and people who had been hospitalised with influenza in 2017 to 2019 ($n$ = 16,058).

- Compared with people in the general population, people who had had a COVID-19 hospitalisation were more than twice as likely to be rehospitalised or die more than a week after discharge, with higher risks overall and for a range of specific causes.

- COVID-19 patients had broadly similar or lower risk of several outcomes compared with influenza patients, but risk of death overall, readmissions or death due to the initial infection, and dementia death were higher in COVID-19 patients.

### What do these findings mean?

- Large numbers of people have been hospitalised with COVID-19 during the pandemic, and the raised risks of death and readmission to hospital that we observed in these individuals could significantly impact public health and resources.

- Risks might be minimised or mitigated by increasing monitoring of patients in the months following hospital discharge, and greater awareness among patients and clinicians of potential problems.

## Introduction

Severe Acute Respiratory Syndrome Coronavirus 2 (SARS-CoV-2) emerged in early 2020 and rapidly spread around the world, infecting >140 million people globally [1]. Acute infection can be asymptomatic or mild [2], but a substantial minority of infected people experience severe Coronavirus Disease 2019 (COVID-19) requiring hospitalisation [3], with age being a major risk factor, along with male sex, non-white ethnicity, and certain comorbidities [4–6]. Early in the pandemic, the proportion surviving hospitalisation was around 50% to 70% [7], though improved treatment guidelines and the identification of effective therapies such as dexamethasone helped to improve survival rates [8,9]. There is now a large and growing population of people who have survived a COVID-19 hospitalisation, but little is known about their longer-term health outcomes.

One recent study of United States Department of Veterans Affairs (VA) data examined a wide range of diagnoses, prescriptions, and laboratory abnormalities among 30-day survivors of COVID-19, showing excess risks of several health outcomes in the 6 months following infection, compared with the general VA population [10]. Whether these findings generalise to the entire US population or other settings remains unclear. Another US study limited to people aged <65 years also found excess risks of a range of clinical outcomes ascertained from health insurance data among people with a record of SARS-CoV-2 infection [11]. A United Kingdom study of routinely collected primary care and hospitalisation data described raised rates of all-cause hospital admission and death among patients discharged following a COVID-19 hospitalisation; the authors also noted raised risks of adverse respiratory and cardiovascular sequelae among the selected outcomes investigated [12]. Only a general population comparator was used, making it difficult to disentangle risks specific to COVID-19 from those associated with hospitalisation more generally; furthermore, a hospitalised cohort is likely to

have been more prone to health problems at the outset than the general population comparator group.

Given high rates of current and past SARS-CoV-2 infection in many countries, understanding risks to health beyond acute infection is vital to support resource planning and inform measures to mitigate and reduce risks. To generate new knowledge and fill gaps in the evidence base in this important emerging area, we therefore aimed to investigate the incidence of subsequent hospital admission and death, both overall and from a wide range of specific causes, following a COVID-19 hospitalisation in England. We aimed to compare post-COVID risks with 2 separate comparison groups: (i) the general population; and (ii) people hospitalised for influenza prior to the current pandemic. The latter was included to provide a comparison with risks after hospitalisation in general, using admissions from a well-characterised infectious disease.

## Methods

### Study design and study population

A cohort study was carried out within the OpenSAFELY platform, which has been described previously [6]. We used routinely collected electronic data from primary care practices using TPP SystmOne software, [13] covering approximately 40% of the population in England, linked at the individual patient level to NHS Secondary Uses Service (SUS) data on hospitalisations, and Office of National Statistics (ONS) death registration data (from 2019 onwards). A brief outline study plan was created in February 2021 setting out a priori the aim and overall approach for the present study (S1 Outline Study Plan); the design was developed further in discussion with the study team prior to data analysis. We selected all individuals discharged between 1 February and 30 December 2020 from a hospitalisation that lasted >1 day and where COVID-19 was coded as the primary diagnosis (based on the International Classification of Diseases (ICD)-10 codes U07.1 "COVID-19—virus identified" and U07.2 "COVID-19 —virus not identified") and who were alive and under follow-up in a TPP practice 1 week after discharge (to avoid a focus on hospital transfers and immediate readmissions/deaths, as early descriptive data suggested a large number of outcomes in week 1 would have obscured the longer-term outcomes that were of primary interest). We excluded a small number of people with missing age, sex, or index of multiple deprivation, which are likely to indicate poor data quality. Two comparison groups were also selected. First, we identified people under follow-up in the general population in 2019, individually matched 5:1 to the COVID-19 group on age (within 3 years), sex, Sustainability and Transformation Plans (STP, a geographical area used as in NHS administration, of which there were 32 in our data), and calendar month (e.g., a patient discharged from a COVID-19 hospitalisation in April 2020 was matched to 5 individuals of the same age, sex, and STP who were under follow-up in general practice on 1 April 2019). The rationale for matching to 2019 data was to provide a comparison with routinely faced risks during prepandemic times. Second, we identified all individuals discharged from hospital in 2017 to 2019 where influenza was coded as the primary reason for hospitalisation and who were alive and under follow-up 1 week after discharge.

### Outcomes and covariates

The outcomes were (i) time to first hospitalisation or death (composite outcome); (ii) all-cause mortality; and (iii) time to first cause-specific hospitalisation or death. Hospitalisations were identified from linked SUS data and included all admissions (whether planned or unplanned). All-cause mortality was identified using date of death in the primary care record so that deaths before 2019 (in the influenza group) could be included (as linked ONS data were not available

prior to 2019); concordance of death dates between primary care and linked ONS data has been shown to be high [14]. Cause-specific outcomes were categorised based on ICD-10 codes into infections (ICD-10 codes beginning with "A"), cancers except nonmelanoma skin cancer (C, except C44), endocrine/nutritional/metabolic (E), mental health and cognitive (F, G30 and X60-84), nervous system (G, except G30), circulatory (I), COVID-19/influenza/pneumonia/ other lower respiratory tract infections (LRTIs) (J09-22, U07.1/2), other respiratory (J23-99), digestive (K), musculoskeletal (M), genitourinary (N), and external causes (S-Y, except X60-84). For each of these, the outcome was time to the earliest of hospitalisation with the relevant outcome listed as primary diagnosis, or death with the relevant outcome listed as the underlying cause on the death certificate [15]. The influenza control group was restricted to those discharged in 2019 for analyses of these cause-specific outcomes, because we did not have linked death registration data (and thus cause of death) for earlier years.

Other covariates considered in the analysis were factors that might be associated with both risk of severe COVID-19 and subsequent outcomes, namely age, sex, ethnicity, obesity, smoking status, index of multiple deprivation quintile (derived from the patient's postcode at lower super output area level), and comorbidities considered potential risk factors for severe COVID-19 outcomes (see Table 1 and footnotes for full specification of covariate categories and comorbidities).

Information on all covariates was obtained by searching TPP SystmOne records for specific coded data, based on a subset of SNOMED-CT mapped to Read version 3 codes. Covariates were identified using data prior to the patient's hospital admission date (for the COVID-19 and influenza groups) or the index date (for the matched control group, i.e., first day of the matched calendar month in 2019). For the COVID-19 and influenza hospitalised groups, primary care data on ethnicity were supplemented with information from the hospitalisation record, to improve completeness. We also classified individuals in residence in a care home based on address linkage; this was used in descriptive and sensitivity analyses only due to limited sensitivity [16]. All codelists, along with detailed information on their compilation are available at https://codelists.opensafely.org for inspection and reuse by the wider research community.

## Statistical analysis

Follow-up began on the eighth day after hospital discharge for the COVID-19 and influenza groups, and on the first of the same calendar month in 2019 for the general population control group. Follow-up ended at the first occurrence of the analysis-specific outcome, or the earliest relevant censoring date for data availability/coverage for the outcome being analysed; the control groups were additionally censored after the maximum follow-up time of the COVID-19 group (315 days). For outcomes involving hospital admissions, the administrative censoring date (for SUS data) was 30 December 2020; for outcomes involving cause of death, the administrative censoring date (for ONS mortality data) was 11 March 2021; for the all-cause death outcome, which was ascertained in primary care data, patients were censored at date of deregistration if they had left the TPP general practice network. Cumulative incidence of the composite hospitalisation/death outcome and all-cause mortality were calculated using Kaplan–Meier methods. Hazard ratios (HRs) comparing COVID-19 and controls were estimated using Cox regression models. Separate models were fitted for the comparisons with matched 2019 general population controls (models stratified by matched set) and with influenza controls (models adjusted for age [continuous, as a 4-knot restricted cubic spline except in cause-specific outcome models where a simpler linear term was used due to lower power], sex, STP, and calendar month). The additional covariates noted above were then added to the models.

**Table 1. Characteristics of patients hospitalised for COVID-19 and controls.**

| | | Hospitalised with COVID-19 | Matched controls from 2019 general population | Hospitalised with influenza in 2017–2019 |
|---|---|---|---|---|
| **N (%)** | | 24,673 (100.0) | 123,362 (100.0) | 16,058 (100.0) |
| **Age (y)** | 18–39 | 2,035 (8.2) | 10,175 (8.2) | 2,024 (12.6) |
| | 40–49 | 2,756 (11.2) | 13,780 (11.2) | 1,462 (9.1) |
| | 50–59 | 4,679 (19.0) | 23,395 (19.0) | 2,126 (13.2) |
| | 60–69 | 4,602 (18.7) | 23,010 (18.7) | 2,653 (16.5) |
| | 70–79 | 5,034 (20.4) | 25,170 (20.4) | 3,492 (21.7) |
| | 80+ | 5,567 (22.6) | 27,832 (22.6) | 4,301 (26.8) |
| | Median (IQR) | 66 (53–78) | 66 (53–78) | 69 (52–80) |
| **Sex** | Male | 13,733 (55.7) | 68,662 (55.7) | 7,097 (44.2) |
| | Female | 10,940 (44.3) | 54,700 (44.3) | 8,961 (55.8) |
| **BMI (kg/m²)** | Not obese | 12,710 (51.5) [54.8] | 82,908 (67.2) [72.7] | 10,065 (62.7) [67.4] |
| N (% of total) | 30–34.9 (Obese class I) | 5,860 (23.8) [25.3] | 20,985 (17.0) [18.4] | 2,853 (17.8) [19.1] |
| [% among nonmissing] | 35–39.9 (Obese class II) | 2,819 (11.4) [12.2] | 7,069 (5.7) [6.2] | 1,271 (7.9) [8.5] |
| | ≥40 (Obese class III) | 1,789 (7.3) [7.7] | 3,015 (2.4) [2.6] | 737 (4.6) [4.9] |
| | *Missing* | 1,495 (6.1) | 9,385 (7.6) | 1,132 (7.0) |
| **Smoking status** | Never | 10,350 (41.9) [42.2] | 52,145 (42.3) [43.0] | 5,711 (35.6) [35.8] |
| N (% of total) | Former | 12,498 (50.7) [51.0] | 52,426 (42.5) [43.2] | 7,346 (45.7) [46.1] |
| [% among nonmissing] | Current | 1,663 (6.7) [6.8] | 16,699 (13.5) [13.8] | 2,874 (17.9) [18.0] |
| | *Missing* | 162 (0.7) | 2,092 (1.7) | 127 (0.8) |
| **Ethnicity** | White | 19,061 (77.3) [78.3] | 80,923 (65.6) [87.7] | 14,035 (87.4) [88.5] |
| N (% of total) | Mixed | 313 (1.3) [1.3] | 821 (0.7) [0.9] | 121 (0.8) [0.8] |
| [% among nonmissing] | South Asian | 3,457 (14.0) [14.2] | 6,727 (5.5) [7.3] | 1,242 (7.7) [7.8] |
| | Black | 920 (3.7) [3.8] | 2,225 (1.8) [2.4] | 251 (1.6) [1.6] |
| | Other | 590 (2.4) [2.4] | 1,572 (1.3) [1.7] | 211 (1.3) [1.3] |
| | *Missing* | 332 (1.3) | 31,094 (25.2) | 198 (1.2) |
| **Index of Multiple Deprivation** | 1 (least deprived) | 4,622 (18.7) | 25,428 (20.6) | 3,282 (20.4) |
| | 2 | 4,743 (19.2) | 25,259 (20.5) | 3,251 (20.2) |
| | 3 | 4,678 (19.0) | 23,503 (19.1) | 3,272 (20.4) |
| | 4 | 5,183 (21.0) | 24,222 (19.6) | 3,133 (19.5) |
| | 5 (most deprived) | 5,447 (22.1) | 24,950 (20.2) | 3,120 (19.4) |
| **Care home resident** | Yes | 1,197 (4.9) | 1,650 (1.3) | 391 (2.4) |
| **Length of hospital stay** | Median (IQR) | 7 (3–13) | - | 4 (2–9) |
| **Any critical care** | Yes | 2,659 (10.8) | - | 18 (0.1) |
| **Comorbidities** | | | | |
| Hypertension | | 12,132 (49.2) | 48,565 (39.4) | 7,550 (47.0) |
| Chronic respiratory disease | | 3,841 (15.6) | 9,664 (7.8) | 3,588 (22.3) |
| Asthma | With no oral steroid use | 3,741 (15.2) | 14,364 (11.6) | 2,872 (17.9) |
| | With oral steroid use | 1,334 (5.4) | 2,375 (1.9) | 1,210 (7.5) |
| Chronic heart disease | | 5,540 (22.5) | 18,285 (14.8) | 3,934 (24.5) |
| Diabetes | With HbA1c <58 mmol/mol | 4,727 (19.2) | 14,855 (12.0) | 2,443 (15.2) |
| | With HbA1c > = 58 mmol/mol | 3,124 (12.7) | 5,567 (4.5) | 1,426 (8.9) |
| | With no recent HbA1c measure | 402 (1.6) | 1,133 (0.9) | 218 (1.4) |
| Cancer (nonhaematological) | Diagnosed <1 year ago | 401 (1.6) | 1,044 (0.8) | 316 (2.0) |

*(Continued)*

**Table 1.** (Continued)

|  |  | Hospitalised with COVID-19 | Matched controls from 2019 general population | Hospitalised with influenza in 2017–2019 |
|---|---|---|---|---|
|  | Diagnosed 1–4.9 years ago | 708 (2.9) | 2,959 (2.4) | 539 (3.4) |
|  | Diagnosed ≥5 years ago | 1,622 (6.6) | 7,353 (6.0) | 1,167 (7.3) |
| Haematological malignancy | Diagnosed <1 year ago | 70 (0.3) | 123 (0.1) | 110 (0.7) |
|  | Diagnosed 1–4.9 years ago | 167 (0.7) | 362 (0.3) | 239 (1.5) |
|  | Diagnosed ≥5 years ago | 252 (1.0) | 694 (0.6) | 295 (1.8) |
| Reduced kidney function | Estimated GFR 30–60 | 4,502 (18.2) | 17,986 (14.6) | 3,299 (20.5) |
|  | Estimated GFR 15-<30 | 481 (1.9) | 1,313 (1.1) | 350 (2.2) |
|  | Estimated GFR <15 or dialysis | 443 (1.8) | 353 (0.3) | 342 (2.1) |
| Chronic liver disease |  | 414 (1.7) | 901 (0.7) | 222 (1.4) |
| Dementia |  | 1,677 (6.8) | 4,409 (3.6) | 1,198 (7.5) |
| Stroke |  | 1,835 (7.4) | 4,275 (3.5) | 1,057 (6.6) |
| Other neurological disease |  | 861 (3.5) | 1,817 (1.5) | 574 (3.6) |
| Organ transplant |  | 173 (0.7) | 168 (0.1) | 189 (1.2) |
| Asplenia |  | 99 (0.4) | 242 (0.2) | 78 (0.5) |
| Rheum arthritis/lupus/ psoriasis |  | 2,132 (8.6) | 7,717 (6.3) | 1,408 (8.8) |
| Other immunosuppressive disease |  | 76 (0.3) | 311 (0.3) | 108 (0.7) |

BMI, body mass index; COVID-19, Coronavirus Disease 2019; GFR, glomerular filtration rate.

Diabetes HbA1c category was determined according to the most recent glycated haemoglobin (HbA1c) recorded in the 15 months prior to the index date; other neurological disease was defined as motor neurone disease, myasthenia gravis, multiple sclerosis, Parkinson disease, cerebral palsy, quadriplegia or hemiplegia, and progressive cerebellar disease; asplenia included splenectomy or a spleen dysfunction, including sickle cell disease; other immunosuppressive conditions was defined as permanent immunodeficiency ever diagnosed, or aplastic anaemia or temporary immunodeficiency recorded within the last year.

Missing ethnicity was handled using multiple imputation (10 imputations) based on a multi-nomial logistic model including all covariates from the substantive models and an indicator for the outcome of interest; a population-calibrated multiple imputation carried out in a previous analysis in this data sources showed minimal nonrandom missingness in ethnicity data (calibration parameters were close to 0), suggesting missing at random to be a reasonable assumption [6]. People with missing data on body mass index (BMI) or smoking were excluded from regression models. Such a "complete case analysis" is valid under the assumption that missingness is conditionally independent of the outcome [17]; while this assumption cannot be verified in the data (because one cannot condition on the missing values themselves), we had no reason to doubt that recording of BMI/smoking in primary care would have been independently associated with the study outcomes; on the other hand, we deemed the missing at random assumption required for multiple imputation to be unlikely to hold for these variables in primary care (e.g., because smokers or those at the extremes of the weight distribution are more likely to have these data recorded). Cumulative incidence of cause-specific hospitalisation/death outcomes were calculated with deaths from other causes treated as a competing risk [18]. HRs for these outcomes were then estimated from a Cox model targeting the cause-specific hazard, with deaths from competing risks censored. Interactions with follow-up time (classified as <30 days, 30 to <90 days, and ≥90 days from hospitalisation [COVID-19/influenza groups] or entry [general population controls]) were examined to investigate whether any increased risk was concentrated in early follow-up and as an implicit check of proportional hazards. We also checked for proportional hazards in adjustment

covariates by testing for a 0 slope in the Schoenfeld residuals for each adjusted model; where there was evidence of nonproportionality, an interaction between follow-up time and any variables with evidence of nonproportional hazards was added to the model as a sensitivity analysis. In a secondary analysis, we fitted Fine and Gray regression models to characterise overall differences in the cumulative incidence of cause-specific outcomes in the presence of competing risks. Further sensitivity analyses included restricting the COVID-19 group to those with a confirmed infection ICD-10 code (U07.1), adjusting for receipt of critical care in hospital (COVID-19 versus influenza comparison only) and adjusting for care home residence.

The study was approved by the Health Research Authority (REC reference 20/LO/0651) and by the LSHTM Ethics Board (ref 21863). An information governance statement is provided in S1 IG Statement. Data management and analysis were carried out in Python version 3.8 and Stata version 16. This study is reported according to the Reporting of Studies Conducted using Observational Routinely-Collected Data (RECORD) guideline (S1 RECORD Checklist).

## Results

We included 24,673 individuals discharged after a COVID-19 hospitalisation, alongside 123,362 matched controls from the 2019 general population, and 16,058 individuals discharged after influenza hospitalisation in 2017 to 2019 (Figs 1 and S1).

At entry, the COVID-19 group had similar age and sex distribution to the general population groups due to matching but had younger median age and were more likely to be male than the influenza group (Table 1). BMI and smoking were 93% to 99% complete in all groups; those with missing data on these variables (who were excluded from later regression modelling) were more likely to be younger, male, and from more deprived areas (S1 Table). Missing ethnicity (which was handled by multiple imputation) was <2% in the COVID-19 and influenza groups but 25% in the matched control group (no hospital-based ethnicity records were available for this group). The COVID-19 group were more likely to be obese, non-white, and less likely to be current smokers than both comparison groups. Preexisting comorbidities were more common in both COVID-19 and influenza-discharged patients than in general population controls. COVID-19 patients had longer median duration of hospital stay and were more likely to have received critical care during their admission than influenza patients.

Numbers of outcome events are shown in S2 Table. Cumulative incidence of subsequent hospital admission or death after study entry in the COVID-19 group was higher than in general population controls but slightly lower than in the influenza group (cumulative incidence at 6 months [for illustration] = 34.8%, 15.2%, and 37.8% in the 3 groups, respectively; fully adjusted hazard ratio (aHR) across all follow-up = 2.22, 2.14 to 2.30 for COVID-19 versus general population [$p < 0.001$]; 0.95, 0.91 to 0.98 for COVID-19 versus influenza [$p$-0.004], cumulative incidence curves over all follow-up shown in Fig 2A, model-specific HRs shown in Fig 3). Cumulative all-cause mortality was higher in the COVID-19 group than in both the general population and influenza groups (7.5%, 1.4%, and 4.9% at 6 months in the 3 groups, respectively; fully aHR = 4.82, 4.48 to 5.19 for COVID-19 versus general population [$p < 0.001$]; 1.74, 1.61 to 1.88 for COVID versus influenza [$p < 0.001$], Figs 2B and 3). To further explore this, causes of death were examined (S3 Table). A substantial proportion of deaths in the COVID-19 group had COVID-19 listed as the underlying cause (500/2,022, 24.7%), while in the influenza group, ≤5 deaths were coded with influenza as the underlying cause.

Cumulative incidences of cause-specific hospital admissions or deaths are shown in Fig 4. After adjustment for matching factors and other covariates, risks of all cause-specific outcomes were substantially higher in COVID-19 groups than in general population controls (Fig 5).

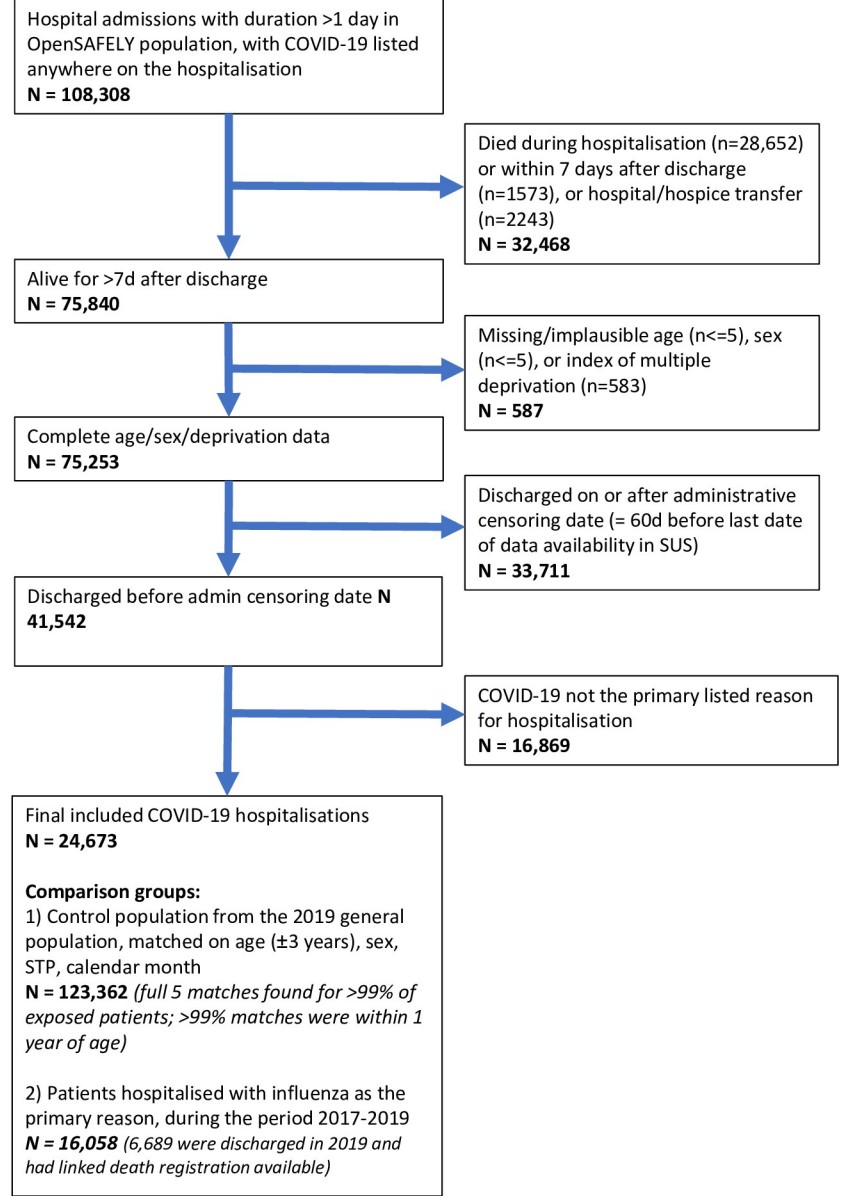

**Fig 1. Study flow chart.** COVID-19, Coronavirus Disease 2019; STP, Sustainability and Transformation Plans; SUS, Secondary Uses Service.

Compared with influenza patients, people in the COVID-19 group had similar or lower risk of admission or death from most causes but higher risks of admission/death from COVID-19/influenza/LRTI (aHR 1.37, 1.22 to 1.54, $p < 0.001$); in the post-COVID-19 group, these outcomes were dominated by codes for COVID-19 itself (515/1,122 [46%] of hospitalisations and 342/368 [93%] of deaths) and pneumonia (461/1,122 [41%] of hospitalisations). The COVID-19 group also had higher risks than the influenza group for mental health or cognitive outcomes (aHR 1.37, 1.02 to 1.84, $p = 0.039$). This was further explored in a post hoc analysis of specific outcomes within the mental health and cognitive category (Table 2). Raised risks in the COVID-19 group appeared to be driven by dementia hospitalisations/deaths (age/sex-adjusted HR 2.32, 1.48 to 3.64, $p < 0.001$), particularly among those with preexisting dementia

(A)

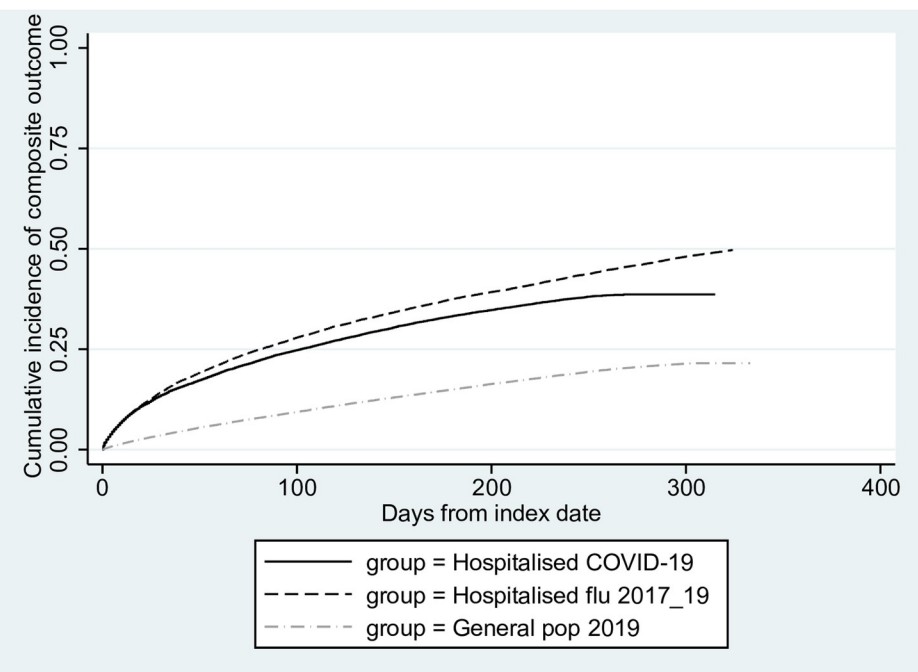

(B)

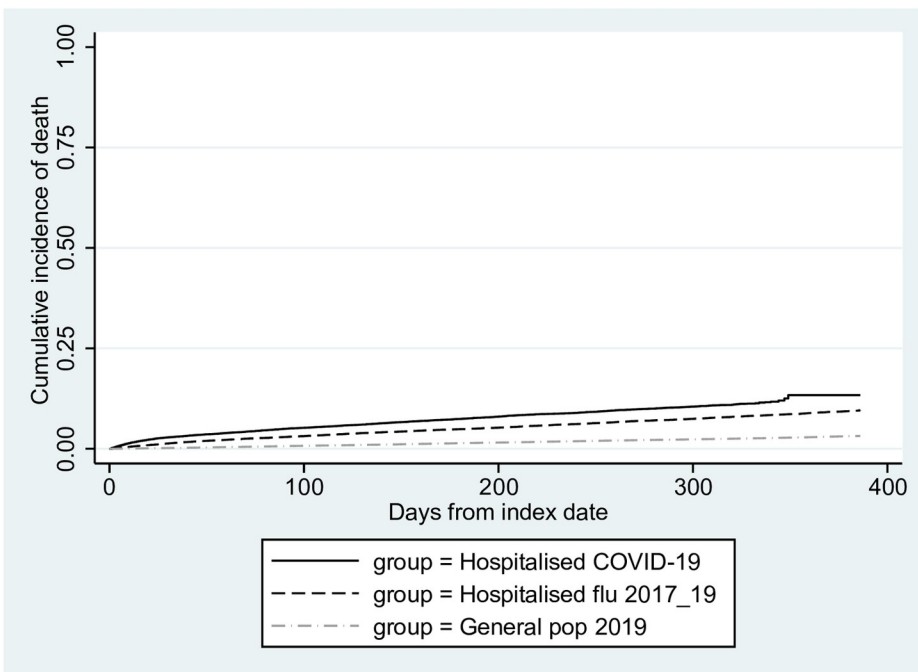

**Fig 2. Cumulative incidence of (A) admission or death (composite outcome), and (B) all-cause mortality, in patients discharged from COVID-19 hospital admissions, influenza hospital admissions, and in matched general population controls.** COVID-19, Coronavirus Disease 2019.

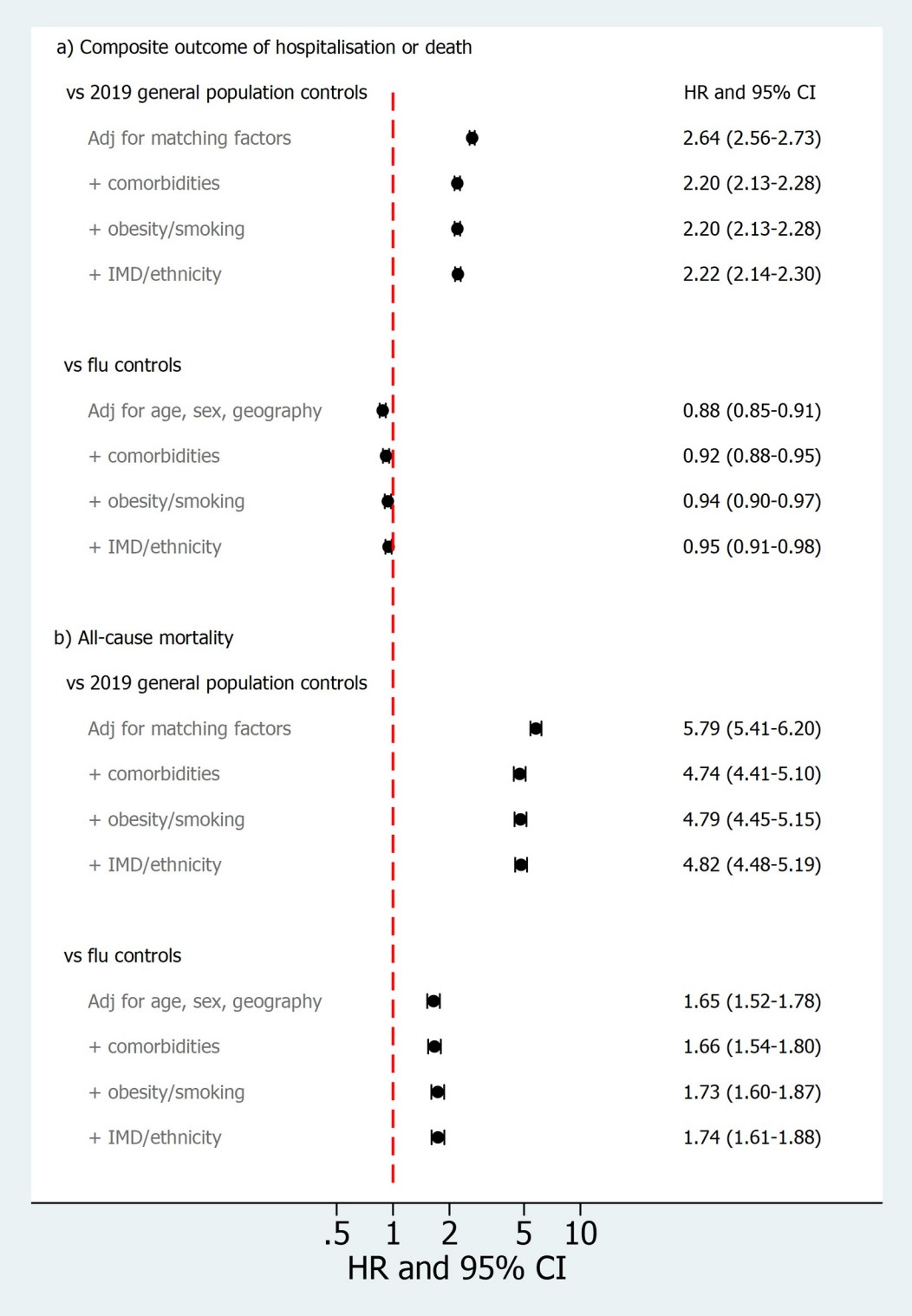

**Fig 3. HRs comparing exposed (prior COVID-19 hospitalisation) and controls for risk of subsequent hospital admission or death (composite outcome) and all-cause mortality.** Footnotes: *All models restricted to individuals with complete data on BMI and smoking (*n* = 23,153/24,673 (94%) in the COVID-19 group, 113,757/123,362 (92%) in general population controls and 14,904/16,058 (93%) in influenza controls (see S1 Table). Median time at risk in the COVID-19 group was 61 days for the composite outcome and 167 days for death; total time at risk followed a bimodal distribution

corresponding to the 2 main pandemic waves in England. BMI, body mass index; COVID-19, Coronavirus Disease 2019; HR, hazard ratio; IMD, index of multiple deprivation.

at baseline (HR 2.47, 1.37 to 4.44, $p$ = 0.002) and/or resident in care homes (HR 2.53, 0.99 to 6.41, $p$ = 0.051). Of note, 129/161 dementia outcome events (80.1%) were deaths (rather than hospitalisations). Higher rates of hospitalisations/deaths due to mood disorders and neurotic/stress-related/somatoform disorders were also observed in COVID-19 patients, but confidence intervals were too wide to be conclusive.

We found evidence of changes over time in the HRs of several outcomes, with more pronounced raised risks earlier following COVID-19 hospitalisation (S2 Fig). Our results changed little in a range of sensitivity analyses, including restricting the COVID-19 group to the 21,770/24,673 (88%) with confirmed infection, adjusting for nonproportional hazards in adjustment variables, for receipt of critical care in hospital, and for care home residence (S3

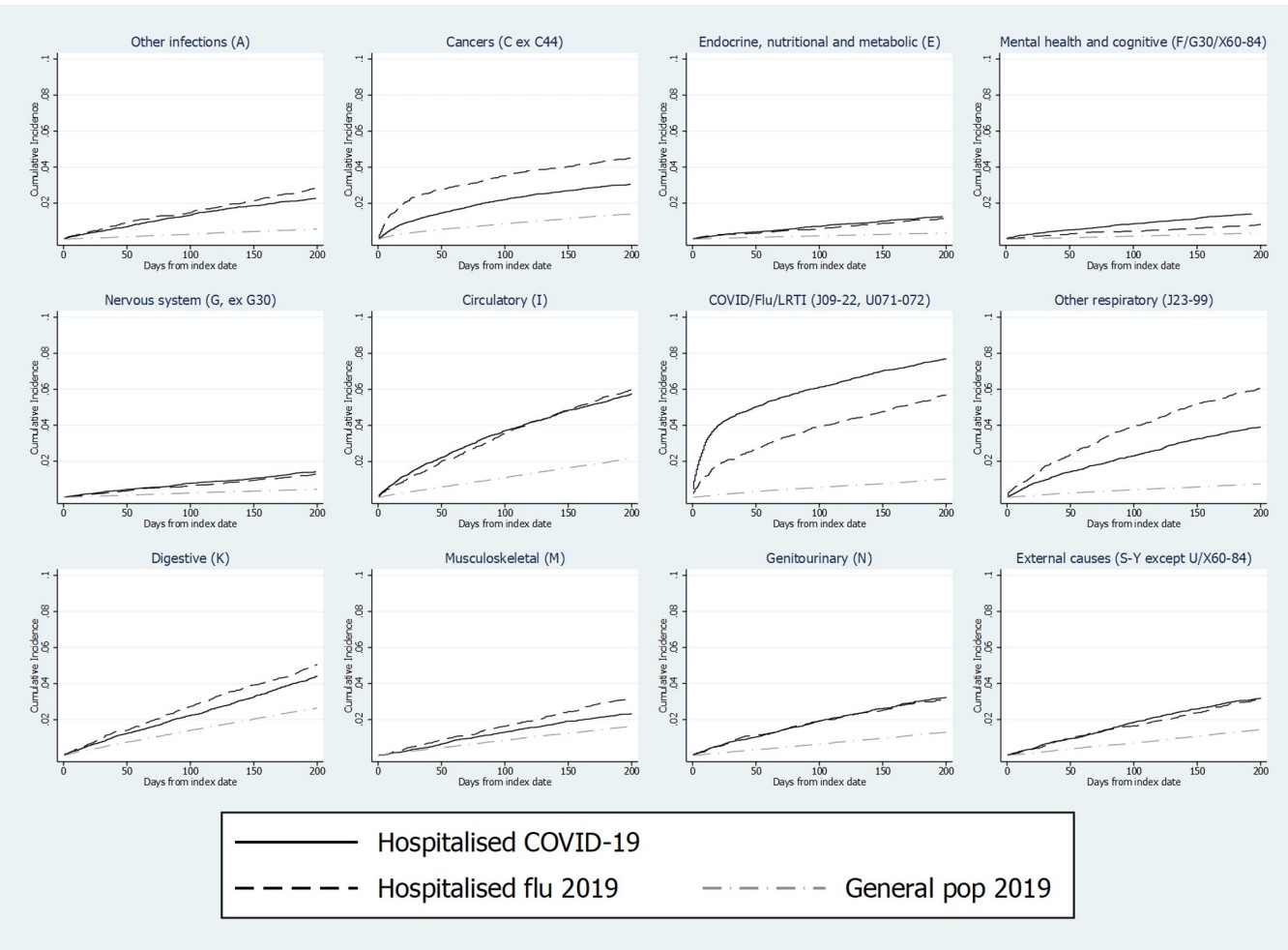

**Fig 4. Cumulative incidence of cause-specific admission/death in patients discharged from COVID-19 hospital admissions, influenza hospital admissions, and in matched general population controls.** Footnotes: For each subpanel, the outcome was defined as the first hospitalisation or death record with an ICD-10 code in the given category listed as the primary reason for hospitalisation/underlying cause of death. Deaths from other causes were treated as competing risks. In the influenza group, only patients entering the study in 2019 were included in analysis of cause-specific outcomes, as linked cause of death data were only available from 2019 onwards. COVID-19, Coronavirus Disease 2019; ICD, International Classification of Diseases; LRTI, lower respiratory tract infection.

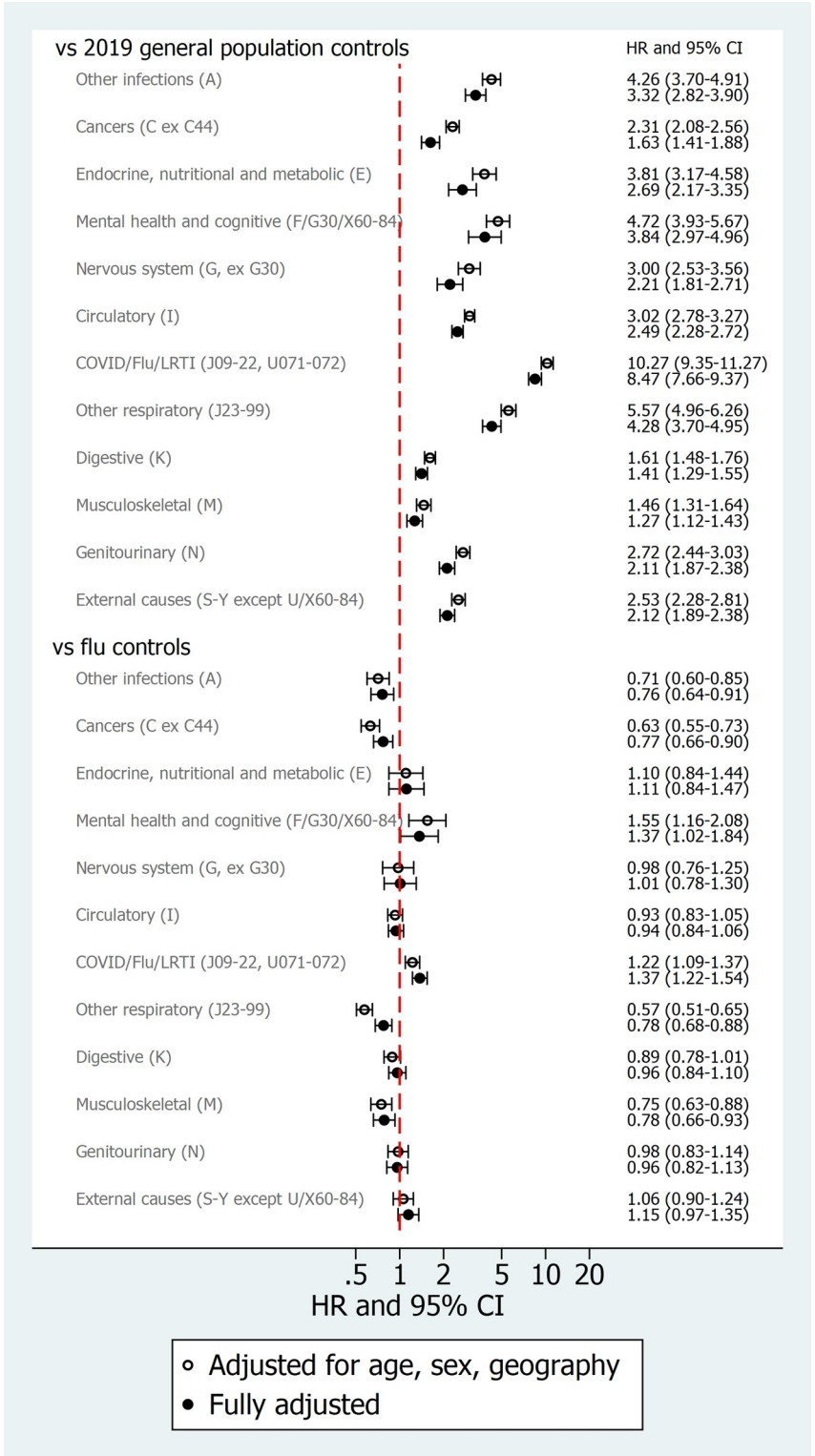

**Fig 5. HRs comparing exposed (prior COVID-19 hospitalisation) and controls for cause-specific hospital admission/deaths.** Footnotes: In the influenza group, only patients entering the study in 2019 were included in analysis of cause-specific outcomes, as linked cause of death data were only available from 2019 onwards. All models restricted to individuals with complete data on BMI and smoking ($n$ = 23,153/24,673 (94%) in the COVID-19 group, 113,757/123,362 (92%) in general population controls and 6,161/6,689 (92%) in influenza (2019 only) controls (see

S1 Table). Median time at risk in the COVID-19 group ranged from 91 to 108 days across outcomes; total time at risk followed a bimodal distribution corresponding to the 2 main pandemic waves in England. BMI, body mass index; COVID-19, Coronavirus Disease 2019; HR, hazard ratio; LRTI, lower respiratory tract infection.

Fig). In secondary analyses using Fine and Gray models, subdistribution HRs were similar to cause-specific HRs from the primary Cox models (S3 Fig).

## Discussion

Patients discharged from a COVID-19 hospitalisation and surviving at least a week had more than double the risk of subsequent hospitalisation or death and a 4.8-fold higher risk of all-cause mortality than controls from the general population, after adjusting for baseline personal and clinical characteristics. Risks were higher for all categories of disease-specific hospital admissions/deaths after a COVID-19 hospitalisation than in general population controls, with excess risks more pronounced earlier in follow-up for several outcomes. Risks for most outcomes were similar or lower for people discharged from a COVID-19 hospitalisation, compared with people discharged from an influenza hospitalisation in 2017 to 2019, but the COVID-19 group had higher subsequent all-cause mortality, higher rates of respiratory infection admissions and deaths (predominantly COVID-19), and more adverse mental health and cognitive outcomes (particularly deaths attributed to dementia among people with preexisting dementia) compared with the influenza group.

Our findings are consistent with emerging evidence from early studies suggesting that a subset of people infected with SARS-CoV-2 can experience health problems for at least several months after the acute phase of their infection, with fatigue, pain, respiratory and cardiovascular symptoms, and mental health and cognitive disturbances being among the problems that have been frequently described under the term "post-acute COVID-19 syndrome" [19]; however, epidemiological characterisation of such sequelae has been limited. Small descriptive studies of COVID-19 survivors have been suggestive of high incidence of a range of outcomes including respiratory, cardiovascular, and mental health related [20,21]; the present study helps to contextualise these observations by adding explicit comparison with risks experienced by the general population and by people with a recent influenza hospitalisation.

**Table 2. Post hoc analysis of specific hospitalisation/mortality outcomes within the mental health and cognitive category.**

| | [events] rate per 1,000 person-years (95% CI) | | Age- and sex-adjusted HR for COVID-19 vs influenza groups (95% CI) |
|---|---|---|---|
| | **COVID-19 group** | **Influenza group** | |
| **Dementia (F00–F03, G30)** | [134] 14.74 (12.45–17.46) | [27] 5.20 (3.57–7.58) | 2.32 (1.48–3.64) |
| *(among those with baseline dementia)* | [102] 168.07 (138.43–204.07) | [14] 59.72 (35.37–100.84) | 2.47 (1.37–4.44) |
| *(among those with no baseline dementia)* | [32] 3.77 (2.67–5.33) | [13] 2.62 (1.52–4.52) | 1.23 (0.63–2.42) |
| *(among those resident in a care home)* | [56] 127.78 (98.34–166.04) | [< = 5] 55.14 (22.95–132.47) | 2.53 (0.99–6.41) |
| *(among those not resident in a care home)* | [78] 9.02 (7.22–11.26) | [22] 4.31 (2.84–6.55) | 1.80 (1.08–2.98) |
| **Delirium (F05)** | [77] 8.47 (6.78–10.59) | [28] 5.39 (3.72–7.81) | 1.10 (0.70–1.72) |
| **Schizophrenia/schizotypal/delusional disorders (F20–29)** | [< = 5] 0.22 (0.06–0.88) | [< = 5] 0.77 (0.29–2.05) | 0.20 (0.03–1.12) |
| **Mood disorders (F30–39)** | [19] 2.09 (1.33–3.28) | [< = 5] 0.77 (0.29–2.05) | 1.61 (0.54–4.79) |
| **Neurotic/stress-related/somatoform disorders (F40–48)** | [13] 1.43 (0.83–2.46) | [< = 5] 0.58 (0.19–1.79) | 2.59 (0.71–9.52) |
| **All except dementia (F05–F99)** | [114] 12.54 (10.44–15.07) | [41] 7.90 (5.82–10.73) | 1.12 (0.77–1.62) |

COVID-19, Coronavirus Disease 2019; HR, hazard ratio.

Only a few other studies to date have compared post-COVID risks with a control group in this way. A recent study of VA data on US veterans examined a wide range of diagnostic and other outcomes in 30-day COVID-19 survivors, compared with the general VA population [10]. Among veterans where COVID-19 had led to a hospitalisation, HRs of every category of outcome were raised. This concurs with findings from our study, despite different characteristics of the VA population. In the UK, an earlier study found an 8-fold higher risk of death in post-acute COVID-19 patients compared with general population controls, and raised risks of respiratory disease, diabetes, and cardiovascular disease [12]. Interestingly, recent data from Denmark suggest limited postacute complications following nonhospitalised COVID-19 [22]; this is in contrast to a recent study using US health insurance data, which found raised risks of a range of outcomes among a relatively young cohort with mostly (92%) nonhospitalised COVID-19 disease, compared with both the general population and people with a record of other viral LRTIs [11].

Our data showed that COVID-19 hospitalised patients were more likely to have baseline comorbidities than general population controls, reflecting known associations between comorbidities and risks of severe COVID-19 outcomes [6]. Differences in outcomes between hospitalised patients and general population controls might therefore reflect baseline differences not fully captured in our adjustment models and might also reflect a generic adverse effect of hospitalisation [23]. This is supported by the more similar risks we observed when COVID-19 survivors were compared with people who had experienced influenza hospitalisation, with risks for some outcomes actually lower in the COVID-19 group, possibly linked to a general reduction in health seeking for non-COVID conditions in the early months of the pandemic [24]. However, all-cause mortality was substantially higher after COVID-19 compared with influenza. A quarter of deaths after a COVID-19 hospitalisation had COVID-19 listed as the underlying cause, but it is not clear from our data whether patients experienced specific complications after hospital discharge that were then attributed to COVID-19, and the possibility of persistent viraemia in these patients cannot be excluded from our data. It is possible that high levels of awareness of COVID-19 during the pandemic may have encouraged coding of subsequent deaths as COVID-19-related, leading to overestimation in the comparison with historical influenza hospitalisations.

Our analysis of cause-specific outcomes also suggested a disproportionate rate of dementia deaths post-COVID-19, particularly among those with preexisting dementia. Cognitive decline after hospitalisation and critical illness have been previously described [25,26]; acute COVID-19 and associated hospital admission, social isolation, and medications may have accelerated progression of patients' dementia; it is unclear whether postdischarge care was adequate for this vulnerable group. However, it is possible that deaths where the underlying cause was recorded as dementia may have been due to progression of underlying health problems following an acute illness as well as difficulty in managing these due to dementia. COVID-19–related delirium may have also triggered or worsened emerging dementia in some patients, or even driven a degree of misclassification given the potential clinical challenge in distinguishing between subacute or chronic delirium and progressive dementia. Due to small numbers, we could not confirm whether higher rates of mood disorders and neurotic/stress-related/somatoform disorders after COVID-19 compared with influenza were due to chance, but a number of previous studies outside the pandemic context have found that critical illness is associated with raised risks of depression, anxiety, and posttraumatic stress [27–29]. It will be important to continue to monitor these outcomes as more follow-up accumulates.

We identified COVID-19 hospitalisations and controls from a base population based on English primary care records. Around 98% of the population are registered with a general practice [30], minimising selection biases due to health-seeking behaviours, and our data

source covered around 40% of the population of England, giving us high statistical power, though it should be noted that our study population would not have been geographically representative of England, since TPP SystmOne software is more widely used than other systems in parts of Eastern and Southern England and used less than other software in London [13]. We examined a broad range of hospitalisation and mortality outcomes and were able to describe and adjust for a wide range of personal and clinical characteristics using rich primary care data. Our findings were robust in a range of sensitivity analyses.

However, our study has some limitations. We relied on ICD-10 codes entered as the primary reason for hospitalisation or underlying cause of death to define our cause-specific outcomes, but these fields may not have been used consistently [31]. In particular, there might have been a tendency for clinicians aware of a recent COVID-19 hospitalisation to code COVID-19 for a range of clinical complications, masking more specific sequelae. Outcomes were classified in broad categories to obtain an overview of post-COVID-19 disease patterns; more granular disease categories would be of future interest but will require more follow-up to maintain statistical power. Our main comparisons may have been affected by time-related factors. We compared post-COVID patients in 2020 with controls from 2019 and earlier; consultations for non-COVID-19 conditions in 2020 are known to have been subdued in the general population [24], perhaps due to lockdown or public reluctance to seek care, potentially affecting comparison with earlier years. On the other hand, patients with a recent COVID-19 hospitalisation may assume immunity from reinfection and be less reticent in seeking care than the general population. The comparison with influenza may also have been affected by seasonality, since the first wave of COVID-19 in England happened outside the typical influenza season. Lack of overlap in the data meant that we could not incorporate seasonal adjustment into our statistical models for this comparison; any confounding by seasonality is likely to have led to underestimation of HRs comparing COVID-19 and influenza patients, since cases of the former were underrepresented in the winter months (which typically confer higher health risks). We had no data on whether influenza hospitalisations were confirmed by PCR testing, raising the possibility of misclassification in this comparator, though we only included cases where influenza was coded as the primary reason for hospitalisation. We did not have detailed data on disease severity, though descriptive data showed that COVID-19 patients tended to have longer hospital stays and more critical care than those hospitalised for influenza. Data were also unavailable on new/emerging COVID-19 variants during the study period. COVID-19 patients in our study had to survive at least a week to enter the study, so our results will not capture the total public health burden from point of discharge given a substantial number of deaths and readmissions in the first week following discharge; however, we felt that excluding this first week enabled a focus on medium and longer-term postacute outcomes and avoided our results being dominated by deaths and readmissions driven by premature discharge and transfers to other hospitals. Our analysis of cause-specific outcomes made an assumption of independent censoring, but deaths from competing outcomes were censored and may have been related to risk of the outcomes under study; our results are likely to have been robust to some violation of independence because the proportion of patients censored due to death from other causes was low (ranging from 1.4% to 2.6% of the study population for specific analyses). Fine and Gray modelling (which does not censor competing events) showed a similar pattern of results to the primary analysis.

Patients surviving a COVID-19 hospitalisation for at least a week after discharge were at substantially higher risk than the general population for a range of subsequent adverse outcomes over a period of up to 10 months' follow-up included in this study. Risks for most outcomes were broadly comparable to those experienced by influenza hospitalisation survivors prior to the pandemic, but in the period following hospital discharge, COVID-19 patients had

higher risks of all-cause mortality, readmission or death attributed to their initial infection, and adverse mental health and cognitive outcomes; in particular, among people with preexisting dementia, we observed an excess of deaths where dementia was recorded as the underlying cause. These findings suggest a need for services to support and closely monitor people following discharge from hospital with COVID-19, for example, through more frequent/active follow-up in primary care in the weeks and months following a hospitalisation. Our results can be used to help inform healthcare providers and raise awareness of potential complications during this period. Our findings will also help with public health resource planning in the context of high rates of SARS-CoV-2 infection in many countries. Ongoing monitoring will be important to investigate whether these patterns persist in the light of new variants and increasing levels of vaccination.

## Patient and public involvement

Patients were not formally involved in developing this specific study design that was developed rapidly in the context of a global health emergency. We have developed a publicly available website (https://opensafely.org/) through which we invite any patient or member of the public to contact us regarding this study or the broader OpenSAFELY project.

The views expressed are those of the authors and not necessarily those of the NIHR, NHS England, Public Health England, or the Department of Health and Social Care.

## Supporting information

**S1 Outline Study Plan. Original priori study plan created in February 2021, and list of changes with justification.**
(PDF)

**S1 IG Statement. Information Governance Statement.**
(PDF)

**S1 Record Checklist. Completed REporting of studies Conducted using ObseRvational Data (RECORD) checklist.**
(PDF)

**S1 Fig. Distribution of entry dates for those in the COVID-19 hospitalised group and the influenza-hospitalised and matched general population comparison groups.** COVID-19, Coronavirus Disease 2019.
(PDF)

**S2 Fig. Changes over time in the HRs comparing outcomes in the COVID-19 and control groups.** COVID-19, Coronavirus Disease 2019; HR, hazard ratio; LRTI, lower respiratory tract infection.
(PDF)

**S3 Fig. aHRs/sHRs in sensitivity analyses.** aHR, adjusted hazard ratio; COVID-19, Coronavirus Disease 2019; HR, hazard ratio; sHR, subdistribution hazard ratio.
(PDF)

**S1 Table. Demographic characteristics of people excluded from complete case analyses due to missing obesity or smoking data.**
(PDF)

**S2 Table. Distribution of first outcomes (hospital admission or death) among included individuals.**
(PDF)

**S3 Table. Leading causes of death in COVID-19 and influenza groups.**
(PDF)

## Author Contributions

**Conceptualization:** Krishnan Bhaskaran, Christopher T. Rentsch, William J. Hulme, Liam Smeeth, Ben Goldacre.

**Data curation:** Krishnan Bhaskaran, Christopher T. Rentsch, William J. Hulme, Helen J. Curtis, Chris J. Bates, Brian MacKenna, Alex J. Walker, Caroline E. Morton, Peter Inglesby, Ian J. Douglas, Helen I. McDonald, Jonathan Cockburn, David Evans, John Parry, Frank Hester, Sam Harper, Sebastian Bacon.

**Formal analysis:** Krishnan Bhaskaran.

**Funding acquisition:** Ben Goldacre.

**Investigation:** Krishnan Bhaskaran.

**Methodology:** Krishnan Bhaskaran, Elizabeth J. Williamson.

**Project administration:** Krishnan Bhaskaran, Helen J. Curtis, Amir Mehrkar, Elizabeth J. Williamson, Liam Smeeth, Ben Goldacre.

**Software:** George Hickman, William J. Hulme, Chris J. Bates, Alex J. Walker, Caroline E. Morton, Peter Inglesby, Jonathan Cockburn, David Evans, Frank Hester, Sebastian Bacon.

**Supervision:** Liam Smeeth, Ben Goldacre.

**Visualization:** Krishnan Bhaskaran.

**Writing – original draft:** Krishnan Bhaskaran.

**Writing – review & editing:** Christopher T. Rentsch, George Hickman, William J. Hulme, Anna Schultze, Helen J. Curtis, Kevin Wing, Charlotte Warren-Gash, Laurie Tomlinson, Chris J. Bates, Rohini Mathur, Brian MacKenna, Viyaasan Mahalingasivam, Angel Wong, Alex J. Walker, Caroline E. Morton, Daniel Grint, Amir Mehrkar, Rosalind M. Eggo, Peter Inglesby, Ian J. Douglas, Helen I. McDonald, Jonathan Cockburn, Elizabeth J. Williamson, David Evans, John Parry, Frank Hester, Sam Harper, Stephen JW Evans, Sebastian Bacon, Liam Smeeth, Ben Goldacre.

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
