## [Editor Report · Decision Letter 0]

15 Jun 2021

Dear Dr Bhaskaran, 

Thank you for submitting your manuscript entitled "Overall and cause-specific hospitalisation and death after COVID-19 hospitalisation in England: cohort study in OpenSAFELY using linked primary care, secondary care and death registration data" for consideration by PLOS Medicine.

Your manuscript has now been evaluated by the PLOS Medicine editorial staff and I am writing to let you know that we would like to send your submission out for external peer review.

However, before we can send your manuscript to reviewers, we need you to complete your submission by providing the metadata that is required for full assessment. To this end, please login to Editorial Manager where you will find the paper in the 'Submissions Needing Revisions' folder on your homepage. Please click 'Revise Submission' from the Action Links and complete all additional questions in the submission questionnaire. Please also include line numbers in your re-submitted manuscript. 

Please re-submit your manuscript within two working days, i.e. by Jun 17 2021 11:59PM.

Kind regards,

Louise Gaynor-Brook, MBBS PhD

Associate Editor

PLOS Medicine

---

## [Decision Letter · Decision Letter 1]

8 Sep 2021

Dear Prof. Bhaskaran,

Thank you very much for submitting your manuscript "Overall and cause-specific hospitalisation and death after COVID-19 hospitalisation in England: cohort study in OpenSAFELY using linked primary care, secondary care and death registration data" (PMEDICINE-D-21-02562R1) for consideration at PLOS Medicine. 

Your paper was evaluated by four independent reviewers, including a statistical reviewer, and was discussed among all the editors here and with an academic editor with relevant expertise. The reviews are appended at the bottom of this email and any accompanying reviewer attachments can be seen via the link below:

[LINK]

In light of these reviews, I am afraid that we will not be able to accept the manuscript for publication in the journal in its current form, but we would like to consider a revised version that addresses the reviewers' and editors' comments. Obviously we cannot make any decision about publication until we have seen the revised manuscript and your response, and we plan to seek re-review by one or more of the reviewers. 

We expect to receive your revised manuscript by Sep 29 2021 11:59PM. Please email us (plosmedicine@plos.org) if you have any questions or concerns.

We look forward to receiving your revised manuscript. 

Sincerely,

Louise Gaynor-Brook, MBBS PhD

Associate Editor 

PLOS Medicine

plosmedicine.org

General comments:

Generally, the style should be "We included 24,673 post-discharge COVID-19 patients ...", for example; numbers should be written out at the start of sentences

Throughout the manuscript, please avoid inadvertent causal language e.g. “associated with higher risk” rather than “COVID-19 patients had higher risk...”

Throughout the paper, please adapt reference call-outs to the following style: "... every year [1,2]." (noting the absence of spaces within the square brackets).

Data availability:

PLOS Medicine requires that the de-identified data underlying the specific results in a published article be made available, without restrictions on access, in a public repository or as Supporting Information at the time of article publication, provided it is legal and ethical to do so. If the data are not freely available, please describe briefly the ethical, legal, or contractual restriction that prevents you from sharing it. Please also include contact information for data request enquiries (web or email address), noting that a study author cannot be the contact person for the data.

Title: Please revise your title according to PLOS Medicine's style. Please place the study design in the subtitle (ie, after a colon). We suggest “Overall and cause-specific hospitalisation and death after COVID-19 hospitalisation in England: a cohort study using linked primary care, secondary care and death registration data in the OpenSAFELY platform” or similar

Abstract Methods and Findings:

Please provide brief demographic details of the study population (e.g. sex, age, ethnicity, etc)

Please include the study design in your Abstract.

Please include the important dependent variables that are adjusted for in the analyses.

Please specify that aHRs are presented with 95% CIs 

Please provide numerical results to substantiate that “COVID-19 survivors with pre-existing dementia had higher risk of dementia death”

In the last sentence of the Abstract Methods and Findings section, please describe 2-3 of the main limitation(s) of the study's methodology.

Abstract Conclusions:

Please begin your Abstract Conclusions with "In this study, we observed ..." or similar, to summarize the main findings from your study without overstating your conclusions. Please address the study implications substantiated by the results. 

Author Summary:

In the final bullet point of ‘What Do These Findings Mean?’, please describe the main limitations of the study in non-technical language.

Introduction:

Please indicate whether your study is novel and how you determined that, being careful to temper assertions of primacy.

Methods:

Did your study have a prospective protocol or analysis plan? Please state this (either way) early in the Methods section. If a prospective analysis plan (from your funding proposal, IRB or other ethics committee submission, study protocol, or other planning document written before analyzing the data) was used in designing the study, please include the relevant prospectively written document with your revised manuscript as a Supporting Information file to be published alongside your study, and cite it in the Methods section. A legend for this file should be included at the end of your manuscript. If no such document exists, please make sure that the Methods section transparently describes when analyses were planned, and if/when reported analyses differed from those that were planned. Changes in the analysis-- including those made in response to peer review comments-- should be identified as such in the Methods section of the paper, with rationale. If a reported analysis was performed based on an interesting but unanticipated pattern in the data, please be clear that the analysis was data-driven.

Please ensure that the study is reported according to the RECORD guideline, and include the completed RECORD checklist as Supporting Information. Please add the following statement, or similar, to the Methods: "This study is reported as per the REporting of studies Conducted using Observational Routinely-collected Data (RECORD) guideline (S1 Checklist)." The RECORD guideline can be found here: https://www.record-statement.org/ When completing the checklist, please use section and paragraph numbers, rather than page numbers which will likely no longer correspond to the appropriate sections after copy-editing.

Results: 

Please incorporate Figure A1 into the main paper. 

Where adjusted analyses are presented, please also provide the unadjusted analyses, and indicate which factors are adjusted for (e.g. please list comorbidities) in the tables/figure legends. 

Discussion:

Please present and organize the Discussion as follows: a short, clear summary of the article's findings; what the study adds to existing research and where and why the results may differ from previous research; strengths and limitations of the study; implications and next steps for research, clinical practice, and/or public policy; one-paragraph conclusion.

Please remove all subheadings within your Discussion e.g. Key findings

Tables:

Please define abbreviations used in the table legend of each table e.g. GFR (Table 1) 

Table A1 - please specify what is represented by numbers in brackets ( )

References:

Please ensure that journal name abbreviations match those found in the National Center for Biotechnology Information (NCBI) databases, and are appropriately formatted and capitalised.

Please also see https://journals.plos.org/plosmedicine/s/submission-guidelines#loc-references for further details on reference formatting. 

Supplementary files: 

Please see https://journals.plos.org/plosmedicine/s/supporting-information for our supporting information guidelines. 

Comments from the reviewers:

Reviewer #1: 

Thanks for the opportunity to review your manuscript. My role is a statistical reviewer so my comments focus on the design, analysis, and data in this manuscript. I've put overall comments first followed by queries relevant to a specific section of the manuscript (with a line reference).

This is an interesting and important study. Routinely collected health data was used to create a cohort that enable a comparison of medium-term outcomes in patients hospitalised for COVID-19, to hospitalised influenza patients and general population (patients registered in the GP system). The COVID patients were directly matched to the comparators based on age (within 3 years), sex, area, and calendar month. Three outcomes were examined - all-cause mortality/first hospitalisations, all-cause mortality, and cause-specific hospitalisation. COVID patients had higher risk of the composite outcome than general population, but similar rates to the influenza in-patients. COVID patients had higher mortality than the other groups, and similar risk of cause-specific apart from subsequent infection or other respiratory infection and mental health/cognitive related admissions/deaths.

The manuscript is well-written and clearly described (particularly given the complexities of assembling and analysing the data). As a side-note the fact that it is possible to do this study so quickly shows the value of setting up the capacity for linked health data to be used in research and is very impressive (I am jealous you have such timely access to this data). 

The GP records were used to create the 'general population group' - is it effectively universal for everyone to be on this database, is there the possibility that this could miss patients who can't/don't seek healthcare? 

L106. The use of the general population comparator is clear, perhaps worth clarifying here in the aims that the influenza hosp. population addresses the issue of potential for increased risk of events after any hospital exposure. 

L114. Is the 40% sample broadly representative? Do some regions contribute more than expected than from their relative population levels? 

L128. What was the rationale of matching to the same calendar date a year earlier rather than the same year? Was this because of availability of linked mortality data? This does make the assumption that overall mortality and hospitalisations rates of those not affected by COVID or influenza were similar in 2019 to 2020 - is there any information you can provide that supports this?

L129. Are the ICD-10 codes used here the same as presented in Table A1? 

L134. Were the hospitalisations used to derive the outcomes 'acute', e.g. would this outcome include in-patient episodes for planned procedures like cataract surgery or just unplanned admissions?

L164. Why was the 8th day after discharge used for COVID and Influenza patients? Also, do influenza patients have similar in-hospital stays to the COVID patients? E.g. time spent in ICU, time on mech ventilation, overall LOS? 

L174. The K-M graphs seem to show proportional hazards for comparisons between the key groups, how was this assumption checked during the analysis for these, and also the other covariates (and the form of any continuous covariates in the analysis)?

L177. How was age included in the model, as a categorical variable or continuous? 

L180. This is certainly true, but there hasn't been any information presented to demonstrate that MAR is a likely mechanism of missingness. The number excluded from each of these models should be presented, and with a comparison of characteristics by missingness. This does also make the interpretation of the difference between the adjusted and non-adjusted models more difficult as they are being done on different samples. In the previous publication using OpenSAFELY, an imputation model was used to handle the missing ethnicity - could this be applied for these analyses as well? 

L185. What software was used for these analyses and data management?

L185. Censoring for death assumes that the cause-specific event would be independent of mortality, which seems unlikely. Typically for this type of analysis I would use a Fine-Grey model rather than K-M, although the interpretation of effect estimates from F-G is little more complicated than K-M. Would you consider that censoring is independent from the hospitalisations here, and did you do any sensitivity analyses to check if the K-M estimator is reasonably here?

L263. Using influenza is clever but it does assume that the hospital stay for influenza and COVID patients is similar - I think this is something that should be acknowledged or some more context provided so it's clear how comparable these patients are. 

Figure A1. There is small cell suppression on the consort flow-chart (second box of exclusions) - was this a condition of the accessing the data? It's not something I'm used to, but I can still make sense of what is happening and reconcile the numbers so I think this is ok.

Figure 1. These figures are ok for review purposes but they could be smartened up a bit, e.g. directly labelling the cum incidence curves instead of the legend would make it easier to read (the legend takes up ~25% of the plot area), maybe colour in addition to line type, label on the Y axis

Figure 2. Again - perfectly readable, but maybe a few tweaks (e.g. x axis goes out to 20 when there doesn't seem to be any CIs that reach that level), also the null effect line overwrites some of the labels.

Reviewer #2: Rosie Cornish

This study provides important evidence about outcomes following hospitalisation due to COVID-19. I only have a few comments. 

1. The authors have used a complete case analysis, which is justified as being appropriate provided missingness is conditionally independent of the outcome. Did the authors investigate this? It seems like a plausible assumption, but it would be helpful to state somewhere whether the data supported this.

2. Care home status is listed in Table 1 but it is not mentioned as a variable included in the fully adjusted models. Since this is a factor strongly associated with COVID-19 as well as the outcomes, I think it should be included as a covariate.

3. In the methods the authors state that COVID-19 discharges were matched on age, sex, geography and time (month). It might therefore be helpful to explain in the text why the distribution of entry dates for COVID-19 discharges does not match the distribution among the general population controls as shown in Figure A2.

4. It would be helpful to indicate on Figures 2 and 4 the numbers included in each analysis so readers can see how the sample sizes for the different models differ.

5. In the results section the authors state that people in the COVID-19 group had similar or lower risk of admission or death from most causes, whereas in the abstract they say that most risks were similar (i.e. "or lower" is omitted). I think it is important to state this in the abstract as well as in the results.

6. Figure 1 states that there were 79,502 matched controls but a figure of 123,362 is given elsewhere.

7. The final point on Figure 1 states "(6,689 were discharged in 2019 and had linked data registration available)". I think this should read "… and had linked death registration data available".

Reviewer #3: This study used routinely collected health administrative data covering 40% of England's population to examine medium to long-term hospitalization and mortality following acute COVID-19. They included 24,673 patients discharged between February and December 2020 who survived 8 days without readmission. Comparison groups were: (i) matched patients in the general population in 2019, and (ii) patients discharged from hospital with influenza (2017-2019). Although incidence of subsequent hospitalisation or death in the COVID-19 group was higher than in general population controls, incidence was comparable to the influenza group (34.8%, 15.2%, and 37.8%, respectively). Compared to the influenza group, the COVID-19 group had higher all-cause 6-month mortality (7.5% vs. 4.9%), and greater risk of death for patients with dementia (HR 2.32). This study demonstrates the importance of selecting appropriate control groups and presents a new finding on the association between COVID-19 and dementia.

This paper was well-written and quite succinct. The topic of long-term outcomes after COVID-19 is highly topical and important given how many people have had COVID-19 infection. The inclusion of a hospitalized influenza control group (from before the pandemic) was appropriate, and consistent with other research (Verma AA et al. CMAJ March 22, 2021 193 (12) E410-E418).The stratified analyses around the dementia outcome is appreciated- this is helpful to address possible confounding related to care home residence. 

Major comments:

The authors should further elaborate on the relevance of the primary research question - the introduction and discussion should make a case for why this study needed to be done, beyond just "to strengthen the evidence base" (line 102). Furthermore, the discussion needs further detail on how the results might be useful, and what further research would be most valuable here. 

Further discussion is also warranted about why COVID patients had more COVID readmissions in the context of published data on long COVID symptoms as well as persistence of viremia in some cases. These are possibly important differences with influenza that explain the findings. 

The findings with respect to dementia are interesting however what is missing is discussion of the relationship between COVID, delirium and dementia. We know that delirium episodes often worsen dementia, and that it is also often difficult to distinguish clinically between subacute/chronic delirium and progressive dementia. Hence it is quite possible that the signal related to dementia outcomes reflects COVID-related delirium (which is well described), either directly or indirectly. The other important consideration is that those who were diagnosed with COVID were isolated (presumably to a greater extent than those with influenza), and a lack of interpersonal contact is also known to worsen dementia/delirium outcomes. 

Other suggestions for improvement:

Introduction

- the middle paragraph has a lot of information about other literature which would be better placed in the discussion- I suggest reducing the amount of detail in here and instead making a stronger case for why this research needed to be done. 

Methods

- The authors point to a previous paper describing the OpenSAFELY program, but I think some more details on this are needed. For example, which 40% of the population does it cover? Is this percent of the population generalizable to the other 60% of the population? I ask this because the overall sample appears low (108,308) compared to the overall number of people who have been diagnosed with COVID-19 in England 

- what does "under follow-up" mean? (ln. 120)- does this just mean they had not yet experienced an outcome? Please clarify. 

Results

- what was the median, and minimum observation time for outcomes across patients? I see that the maximum was 315 days. This question also relates to what "under follow-up" means- how complete was the follow-up? Since this sounds like health administrative data, my expectation was that one would only be lost to follow-up if they moved away, but this should be clarified. 

- tables and figures were very helpful and nicely done

- I note that the COVID group experienced fewer cancer-related outcomes than the influenza group, however this was not mentioned in the results text. I think this finding is worthy of some discussion (for discussion section as well)- could this be because the COVID group had lower rates of cancer at baseline (possibly due to extra precautions taken by cancer patients to reduce their exposure?). 

Reviewer #4: "Overall and cause-specific hospitalization and death after COVID-19 hospitalization in England: cohort study in OpenSAFELY using linked primary care, secondary care and death registration data" (manuscript ID: PMEDICINE-D-21-02562R1)

SUMMARY: This cohort study of over 164,000 hospitalized adults (n=24,673 discharged following COVID-19 between discharged between 1st February and 30th December 2020; n=123,362 matched general population controls in 2019; and n=16,058 discharged following influenza between 2017-2019) compares the medium and long-term risks of hospital admission and death, overall and by specific cause across the three study groups. It uses administrative data sources from linked primary care and hospital data in the OpenSAFELY platform, which is reported to cover approximately 40% of the population in England. The main finding identifies that people discharged following hospitalization for COVID-19 had higher associated risks for rehospitalization and death than the general population, similar risks compared to those hospitalized for influenza. 

The paper is well written and addresses an evolving, poorly studied, and important area of health policy and planning as it relates to the care of patients who survive hospitalization for COVID-19. Early data indicate these individuals appear to be at high risk for ongoing health needs and high resource use. The present study appears to build on a recent study in the UK (Ayoubkhani D et al BMJ 2021) by including an "active control" population of adults hospitalized with influenza and with a longer study follow-up. The study has clear applications to healthcare resource planning and policy in the care of these individuals suggesting a substantial extra burden on healthcare systems in the future. 

I have a few concerns with the study as it stands that I believe should be addressed to improve the overall quality of this paper, which I hope are viewed as helpful.

1) The authors used Cox regression models to estimate hazard ratios for the defined outcomes. However, a key assumption of this modelling approach is that these HRs are constant over time. Prior literature on risk of death or rehospitalization demonstrates higher risk in the early period following discharge from hospital, as does Figure 1a provided by the authors. It would be helpful if the authors included time as an interaction term in their modelling to help delineate how these risks change over time, including the possibility of categorizing the time variable into clinically relevant post-discharge time frames (e.g. 30-, 90-, 180-day readmission). 

2) Why did the authors choose not to adjust for admission to ICU in their modelling as a surrogate for disease severity? There were nearly 11% of COVID-19 patients and only 0.1% of influenza patients admitted to ICU. This may serve to substantially overestimate the adjusted risks of mortality among patients with COVID-19. Further focus on this could be added to the limitations in the discussion section as well. 

3) Related to #2 above, why did the authors not adjust for readmission risk using one of the many available readmission risk indices (see Kansagara D et al JAMA 2011 for a comprehensive list)?

4) The cause-specific outcomes among adults with COVID-19 may be artificially higher than those with Influenza due to availability bias. Put simply, patients and providers may be much more aware of COVID-19 and its complications, including those related to return to hospital than might be the case for those with pneumonia or even confirmed influenza. This appears to be supported by the data presented in this study. For example, the authors report that, "A substantial proportion of deaths in the COVID-19 group had COVID-19 listed as the underlying cause (500/2022, 24.7%), while in the influenza group ≤5 deaths were coded with influenza as the underlying cause"; and "Compared with influenza patients, people in the COVID-19 group had similar or lower risk of admission or death from most causes, but higher risks of admission/death from COVID-19/influenza/lower respiratory tract infection (LRTI, adjusted HR 1.37, 1.22-1.54); in the post-COVID-19 group these outcomes were dominated by codes for COVID-19 itself (515/1122 [46%] of hospitalizations and 342/368 [93%] of deaths)". I worry that the conclusions drawn from this study in relation to the influenza comparator group may be different when considering the lack of adjustment for disease severity and this potential source of bias. 

5) It is unclear as to why the authors chose to report cumulative incidence of the outcomes at 6 months when the maximum duration of follow-up for COVID-19 patients is 315 days (and the general population group was censored using this time frame). Is this the median follow-up time for the cohort? Further clarification as to why this specific time point was chosen would be informative.

6) The authors compare baseline characteristics of the three study groups and report for example that, "The COVID-19 group were more likely to be obese, non-White and less likely to be current smokers than both comparison groups". It would be helpful to report standardized differences in Table 1 to compare the magnitude of differences between study groups, which may inform some of the residual confounding that may be unaccounted for. 

7) There is a risk of misclassification of hospitalization for COVID using ICD-10 code U07.2 "COVID-19 - virus not identified". To test the robustness of their findings, could the authors repeat a sensitivity analysis among those with only the ICD-10 code "U07.1 COVID-19 - virus identified"? 

8) I am concerned about the presence of immortal time bias in requiring that adults survive 7 days post discharge, which also serves to underestimate the magnitude of healthcare resource needs among adults who survive hospitalization for COVID-19. There were ~3,700 patients who died within 7 days of discharge or were transferred to hospice (where they presumably died), which is ~15% of the final cohort of adults hospitalized with COVID-19 (n=24,673). The authors should mention this in their limitations and modify the interpretation of the study to include that this applies to adults who survive at least 1 week following hospitalization for COVID-19. 

8) How confident are the authors in the accuracy of hospital coding for influenza admissions? Are these hospital coding practices based on PCR confirmed influenza?

9) Can the authors comment on the generalizability of the study in using the OpenSAFELY data? Specifically, there are considerations about the generalizability to the overall study cohort, and to those that acquired COVID. The low proportion of patients residing in care homes (4.9% in the COVID population and 1.3% in the matched general population) may suggest this is a healthier, less frail population overall.

10) I believe the paper could be improved with a more substantive discussion of the policy applications for health and human resource planning in the care of patients who survive hospitalization for COVID-19 beyond, "These findings suggest a need for services to support and closely monitor people following discharge from hospital with COVID-19." For example, how might these patients require more support? Will it require more frequent follow-up from their General Practitioners? Does the healthcare system need to provide education to healthcare providers to improve their comfort and competency to deal with the potentially unique needs of these patients? Should governments consider financial motivations to incentives the care of this vulnerable population?

[LINK]

---

## [Decision Letter · Decision Letter 2]

8 Nov 2021

Dear Dr. Bhaskaran,

Thank you very much for re-submitting your manuscript "Overall and cause-specific hospitalisation and death after COVID-19 hospitalisation in England: a cohort study using linked primary care, secondary care and death registration data in the OpenSAFELY platform" (PMEDICINE-D-21-02562R2) for consideration at PLOS Medicine.

I have discussed the paper with our academic editor and it was also seen again by three reviewers. I am pleased to tell you that, provided the remaining editorial and production issues are fully dealt with, we expect to be able to accept the paper for publication in the journal.

[LINK]

Please let me know if you have any questions, and we look forward to receiving the revised manuscript.   

Sincerely,

Richard Turner PhD, for Louise Gaynor-Brook, MBBS PhD

rturner@plos.org

Requests from Editors:

Please trim the data access statement. If OpenSAFELY does not currently permit access consistent with PLOS' data policy (https://journals.plos.org/plosmedicine/s/data-availability) please state that and briefly explain the reason(s). 

At line 47, please make that "February to December". 

At line 53, for example, please use the general style "13,733"). 

At line 54, please make that "adjusted Hazard Ratio [aHR] 2.22 ..." and "aHR" can be used thereafter in the abstract.

At line 55, should that be "aHR 0.95"?

At line 70, please make that "readmission or death" or similar.

In the abstract and throughout the text, please quote p values alongside 95% CI, where available. 

At line 307 and any other instances, please avoid "nearly 5-fold" in favour of quoting the actual value. 

Please remove the information on data availability, funding and competing interests from the end of the main text. In the event of publication, this information will appear in the article metadata, via entries in the submission form. 

In the reference list, please correct the citations for references 10 and 11. 

Please ensure that journal names are abbreviated consistently (e.g., "Lancet" and derivatives rather than "The Lancet" etc).

We suggest breaking the analysis plan out into a separate attached file labelled "S1_Analysis_Plan" or similar, referred to in the Methods section (main text).

Please adapt the label for the RECORD checklist to "S2_RECORD_Checklist" or similar, and refer to it by this name in the Methods section. 

Comments from Reviewers:

*** Reviewer #1: 

Thanks for the revised manuscript and replies to my original queries. The responses are comprehensive and apart from one small detail (in the flow diagram) I think this manuscript looks excellent and I recommend that it is accepted. I appreciate the work you have done in this revision - it makes the entire body of work robust. 

The GP coverage information is helpful - I think acknowledging this as a limitation is all that's needed here. The additional details and tests about the proportional hazards are appropriate. The additional details and modified analysis (i.e. the ethnicity adjusted models) are appropriate and meet my initial query. The sensitivity analyses using Fine and Grey are a good addition and it's reassuring to see similar results to the Cox models. 

Only one small check for Figure 1 - Is the number in the flowchart that died during hospitalisation or transfer (n=30,070) correct? This doesn't match the number before and after this exclusion was applied (108,308/75840).

*** Reviewer #3: 

Thank you for addressing my comments. 

My only remaining concern relates to the dataset coverage of "OpenSafely": in addition to mentioning data coverage as a limitation in the discussion section, the details of which 40% of the population are covered should be included in the Methods section as well. 

We identified a few minor typographical errors in the "Why was this study done" section- line 86 "to compared", line 100 "by increased". 

*** Reviewer #4: 

The authors should be commended for addressing all of our concerns raised during review, including completing several additional analyses to test the robustness of their findings and adding an expanded discussion of the study limitations and policy applications. As a result, we believe that these steps served to substantially strengthen their conclusions which greatly improved the overall quality of the manuscript. Congratulations on completing this important work!

***

[LINK]

---

## [Editor Report · Decision Letter 3]

17 Nov 2021

Dear Dr Bhaskaran, 

On behalf of my colleagues and our Academic Editor, Dr Basu, I am pleased to inform you that we have agreed to publish your manuscript "Overall and cause-specific hospitalisation and death after COVID-19 hospitalisation in England: a cohort study using linked primary care, secondary care and death registration data in the OpenSAFELY platform" (PMEDICINE-D-21-02562R3) in PLOS Medicine.

Prior to final acceptance, please:

Adapt "hospitalisation/death", and similar forms of words in the abstract, to "hospitalisation or death" as appropriate;

Remove "nearly" at line 308; and 

Trim the section on "Information and governance" to no more than 5 lines, and move it to the Methods section (some of the information is already present in that section and does not need to be duplicated). 

PRESS

Sincerely, 

Richard Turner PhD, for Louise Gaynor-Brook, MBBS PhD 

rturner@plos.org